# Prosthetic Joint Infections: Biofilm Formation, Management, and the Potential of Mesoporous Bioactive Glass as a New Treatment Option

**DOI:** 10.3390/pharmaceutics15051401

**Published:** 2023-05-03

**Authors:** Dana Almasri, Yaser Dahman

**Affiliations:** Department of Chemical Engineering, Toronto Metropolitan University, Toronto, ON M5B 2K3, Canada

**Keywords:** prosthetic joint infection, bone cements, PMMA, calcium sulfate, bioactive glass, mesoporous bioactive glass

## Abstract

Infection of prosthetic joints is one of the biggest challenges to a successful replacement of the joint after a total joint arthroplasty. Such infections are caused by bacterial colonies that are difficult to treat by systemic delivery of antibiotics. Local delivery of antibiotics can prove to be the solution to such a devastating outcome that impacts patients’ health and ability to regain function in their joints as well as costs the healthcare system millions of dollars every year. This review will discuss prosthetic joint infections in detail with a focus on the development, management, and diagnosis of the infections. Surgeons often opt to use polymethacrylate cement locally to deliver antibiotics; however, due to the rapid release of antibiotics, non-biodegradability, and high chance of reinfection, the search for alternatives is in high demand. One of the most researched alternatives to current treatments is the use of biodegradable and highly compatible bioactive glass. The novelty of this review lies in its focus on mesoporous bioactive glass as a potential alternative to current treatments for prosthetic joint infection. Mesoporous bioactive glass is the focus of this review because it has a higher capacity to deliver biomolecules, stimulate bone growth, and treat infections after prosthetic joint replacement surgeries. The review also examines different synthesis methods, compositions, and properties of mesoporous bioactive glass, highlighting its potential as a biomaterial for the treatment of joint infections.

## 1. Introduction

### 1.1. Background on Prosthetic Joint Infection

The number of total joint arthroplasties (TJAs) has been increasing steadily over the years, driven by the demands of the aging population. As life expectancy increases, the demands for better mobility and easier movement may require replacement of joints with implants that can keep up with an active aging population [1,2]. As the number of TJAs increases, the problems associated with these artificial joints also increase. Prosthetic joint infections (PJIs) are one of the most common problems associated with TJAs, since they affect 1% to 2% of all primary arthroplasties and up to 4% of revision arthroplasties [1,2,3]. The high number of TJAs makes prosthetic joint infections very costly for the health care systems, as these infections can lead to implant failure, revision surgeries, loosening of implants, amputations, and other problems [2,3]. Treatment and management of PJIs can be challenging, as it requires several steps that include multiple revision surgeries, long-term administration of antibiotics, and insertion of a new prosthesis [3]. These can have serious consequences on the health of the patient, as multiple surgeries can lead to disability or poor functionality of the joint [1]. Long-term administration of antibiotics can also lead to many problems such as microbial dysbiosis [3] and resistance to antibiotics, making the patients more susceptible to highly resistant bacteria [4]. 

As the annual number of new implants increases, so does the rate of infections. The incident of PJIs increases with time, as more arthroplasty surgeries being performed and more infections are being reported. The economic burden of PJIs is also increasing; PJIs cost the healthcare system around $600 million in 2009 in the US, and the projections indicate that by 2020, their annual cost will increase to around $1.6 billion dollars [5]. It was reported that the revision surgery following an infection is 4.8 times more expensive than the primary arthroplasty of the hip in the United States [6]. These numbers reflect the costs associated with the surgery alone without other indirect costs related to revision surgeries and patient care outside the hospital, as well as the repeated infections of the implant [7]. Revision surgeries that involve implant infections are more costly, as they need full implant replacement, more time to complete the surgery, and there are also additional costs associated with blood loss and bone allografts as well as prolonged antibiotics administration [5]. Additionally, revision surgeries involving infections cost three to six times the cost of primary implantation [5]. 

Implant infection elicits an inflammatory response that leads to the recruitment of neutrophils to the infection site. An innate immune response follows, as phagocytes and macrophages follow neutrophils to the infection site [8]. This causes an increase in white blood cell count, C-reactive proteins, and erythrocytes. Inflammation caused by the infection will manifest in redness in the infection area, pain, swelling, and heat [9]. Bacteria will produce toxins that will lead to the stimulation of osteoclasts, which will act to resorb the bone, and this will lead to the loosening of the implant, which is another painful sign of bacterial infection [8,10]. The implant site is not innervated by blood vessels, which means that the immune system cannot directly attack the bacteria adhering to the implant, making the host defense less effective in combating bacterial infections on implants [9]. The host defense system has limited access to the biofilm as it starts to grow on the implant leading to the failure of polymorphonuclear leukocytes to successfully phagocytize the bacteria, which frustrates the immune response, rendering it ineffective in fighting off the infection [11].

Another factor contributing to the development on the prosthesis is the type of material that it is made of. There are several different metals that can be used to make prostheses for joint replacement. These include titanium, cobalt-based alloys, and stainless steel. Stainless steel is used when bone tissue erodes or is worn out, as they are resistant to corrosion [12]. Alloys are nonmagnetic, heat-resistant, and corrosion-resistant. Lastly, titanium is the prized material used because it is highly durable, resistant to loads, lightweight, and has great compatibility with the human body [12]. Materials used for the implant is another factor contributing to the severity of the infection. Kates et al. (2016) suggested that using titanium rather than stainless steel could reduce the chances of infection. They looked at the infection rate of different materials in vitro and found that there is no big difference between the two; stainless steel has an infection rate of 82%, while titanium has an infection rate of 59%. However, these studies do not reflect the infection rate in vivo, as the tissues surrounding and attached to the implant could affect the infection rate and the exposure to pathogens during or after surgery [9].

Prosthetic joint infections are usually classified based on the time it takes for an infection to start. Any infection that occurs within the first three months of the surgery is considered a primary infection. It is likely to be caused by intraoperative contamination and is usually caused by virulent organisms. Infections that occur after the third month and before 12 months are also caused by intraoperative contamination, but they are likely caused by a less virulent organism [5]. Any infection that occurs after that is most likely caused by hematogenous infection. Tande et al. (2014) examined the types of bacteria that are more likely to cause an infection and found that Gram-positive cocci are involved in most infections with *Staphylococcus aureus* and coagulase-negative staphylococci contributing the most to these infections, as they cause about 50% of all PJI infections [5]. Other cocci bacteria such as streptococci and enterococci cause around 10% of all infections. In a retrospective study conducted by Linke et al. (2022), they evaluated culture-positive synovial samples of different patients. Their results led to the conclusion that the most detected species causing prosthetic joint infections are Staphylococcal species causing around 50% of the infections, and coagulase-negative staphylococci causing around 33% of the PJIs [13]. This is in line with other studies that looked at the pathogens causing these infections. It is important to identify the pathogen as it helps in selecting the most appropriate antibiotics and a debridement procedure to manage the infection [5].

### 1.2. Biofilm Formation and Challenges in Treating Biofilm Infections

Prosthetic joint infections occur when the implants become colonized by different species of bacteria that form a biofilm on the implant. PJIs are hard to diagnose, as bacteria can persist after treatment, and these infections have a high rate of recurrence. Biofilm is defined by Costerton et al. (1999) as a “structured community of bacterial cells enclosed in a self-produced polymeric matrix adhert to an inert or living surface” [14]. Bacterial biofilms often consist of multiple species that share the polymeric matrix, nutrients, and DNA. The founder species prepare the conditions necessary to allow other species to develop and create complex biofilms that can protect themselves from harmful outside interference. Biofilms are dynamic colonies that have cycles of growth, attachment, maturation, and dispersal, as shown in Figure 1 [15]. 

Each part of the biofilm contributes to its survival and persistence. The biofilm is surrounded by extracellular polymeric matrix (EPS), which protects the biofilm from different environmental stresses such as lack of nutrition, dehydration, and water flow. 

The EPS consists of proteins, polysaccharides, and extracellular DNA with concentration and compositions varying according to the organisms in the biofilm [5]. For the biofilm to survive, some cells become dormant after maturation, which is a state where the bacteria are persistent and resistant to different antibiotics. These dormant cells can survive treatments and can rebuild biofilms and re-infect implants even after treatment [9]. 

The presence of EPS creates a physical barrier that prevents host defenses and any antibiotic treatment from affecting the biofilm. It does not allow antibiotics or any antimicrobial to diffuse into the biofilm and kill the bacteria living inside. In fact, the EPS can create acidic or anoxic areas that can degrade the antibiotics before they can kill any bacteria [10]. Secreted polymers on the surface of the biofilms can also bind and sink antibiotics, which is another mechanism of defense used to increase the resistance of biofilms to antibiotics [11]. The limited capacity of the antibiotics to diffuse through the biofilm will reduce the concentration of antibiotics entering the biofilm, thus exposing the bacteria to low doses of antibiotics. When bacteria inside the biofilm are exposed to low concentrations of the antibiotics, they will develop resistance to the antibiotics, which will make the treatment of the biofilm more challenging [11]. 

The host has an innate immune response that can fight off infections in early stages. However, it is often limited when fighting infections on implants as phagocytes and leukocytes, deployed by the body to fight infection, have poor access to the implant surface, rendering them ineffective in fighting the bacteria prior to the development of a biofilm. Once the biofilm matures, the EPS surrounding the biofilm does not allow phagocytes or leukocytes to attack the bacteria inside, which means that the body’s immune system becomes exhausted trying to fight off the infection [9]. The antibodies developed by the immune system can normally fight any bacterial infection or biofilms in the early stages of development, but once the biofilm matures, these cells are ineffective in fighting off infections. The body’s response to biofilm elicits a prolonged inflammation in the tissue that causes tissue damage, which releases nutrients from the damaged cells. These released nutrients are taken up by the biofilm thus aggravating the problem and creating stronger biofilms [11].

### 1.3. Challenges of Biofilm Infection Treatment

Biofilm infections are one of the most difficult infections to treat because of their ability to persist after treatment. These infections are a thousand times more resistant to commonly used antibiotics [11,16,17]. The extracellular matrix surrounding the biofilm protects it from damage that may be caused by antimicrobial and other drugs targeting the biofilm. Biofilms are currently treated by exposing the area of infection to high concentrations of antibiotics for a long period of time. Specifically, it is important to maintain antibiotics at concentrations higher than the minimum effective concentration (MEC) and less than the minimum toxic concentration (MTC) [16]. Some of the factors that affect the capacity of antibiotics to treat biofilm infections include the fact that these antibiotics cannot penetrate the extracellular matrix. For example, some antibiotics such as aminoglycosides are positively charged, while the extracellular matrix of the biofilm is negatively charged, which means that these antibiotics cannot successfully enter the biofilm. Another factor that contributes to resistance is that bacteria inside the biofilm have a different phenotype that encourages slow growth and reduced uptake of different nutrients and antibiotics, so if the antibiotics are not taken up by the bacteria, they will have reduced effectiveness. The third factor that contributes to the ineffectiveness of antibiotics is that enzymes within the biofilm can de-activate antimicrobial activity [16]. 

The failure of antibiotics to penetrate the biofilm means that bacteria on the surface might be exposed to lethal dosage of antimicrobials but the bacteria in the center will only be exposed to sub-inhibitory concentrations [17]. The effectiveness of the antibiotic will also depend on the type of bacteria that form the biofilm and the type of antibiotic used. Studies show that some antibiotics are more effective than others in penetrating the biofilm [17,18,19]. The bacteria strain influences the effectiveness of the antibiotics and antimicrobials used to treat biofilm infections. Some strains of *S. aureus,* for example, can resist antibiotics and stop the antibiotics from penetrating the biofilm, even when the antibiotics are very effective against different strains. Siala et al. (2014) showed that the penetration rate of daptomycin of different strains of *S. aureus* had a wide range of 0.6–52%. The more resistant strains had a lower penetration rate, while the more antibiotic-susceptible strains had a high penetration rate [17,18,19,20]. It is important to consider the strain of bacteria that forms the biofilm, and the different effects that different antibiotics will have on the biofilm. The choice of antibiotic, method of administration, and the drug delivery system are crucial aspects of successfully treating and removing biofilms. 

Biofilm infections need to be treated by an effective drug delivery system that can maintain prolonged drug release. The drug delivery system must deliver antimicrobials at concentrations that are effective and above the minimum inhibitory concentration (MIC) for the treatment period. The drug delivery system needs to have the beginning and the endpoint of treatment to assess its effectiveness in treating biofilm infections [21]. The beginning of treatment usually starts with identifying the offending organisms and determining the most appropriate antimicrobial to be used for treatment. The local concentration of antibiotics is usually determined based on the type of biofilm and its location [21]. It is usually recommended to administer high concentrations of antibiotics for a long period of time to treat a biofilm infection. The concentrations and duration of antimicrobial treatment needed to eradicate biofilm infections are hard to define, as each biofilm infection is caused by different organisms and has different defense mechanisms [22]. Generally, the minimum biofilm eradication concentration (MBEC), which is greater than the MIC by two or more orders of magnitude [22], is what treatments hope to achieve. The MBEC is the lowest level of antimicrobials that can kill bacteria in biofilms. Although the determination of the MBEC is a great challenge for researchers, some estimate the concentration of antibiotics and the duration of exposure based on in vitro studies. Badha et al. (2019) tested the capacity of tobramycin and vancomycin to eradicate biofilms established in rabbit muscle and bone specimens in vitro [22]. Their study showed that while achieving the desired concentration to eradicate biofilm infection is important, it is also important to consider the duration of exposure, which can help to eradicate biofilms at less than toxic levels [22]. 

Jacqueline et al. (2014) suggested that the perfect antibiofilm antibiotic does not exist. They described in their study the ideal antibiotic that can completely eradicate the biofilm through a bactericidal mode of action, by acting against bacteria in the stationary phase, effectively killing pathogens that develop in the biofilms and penetrating through the “slime” [23]. The MBEC of antimicrobials commonly used to treat biofilms is about 10-fold to 8000-fold higher than the MIC, which means that at these concentrations, the antibiotics will be toxic to the patient. They also suggest that the most effective way to eliminate biofilm infections is through combination therapy, where two or more antibiotics are used at the same time. Vergidis et al. (2010) discussed the effectiveness of using more than one antibiotic through in vivo studies, where they used rat models with inserted infected titanium wires and different combinations of antibiotics to treat the infection [24]. They found that combination therapy with the antibiotic of choice linezolid with rifampin or vancomycin was more effective than monotherapy with either antibiotic [24]. When designing a drug delivery system, it is crucial to keep this in mind in order to ensure that the system designed can carry multiple antibiotics and deliver them to the site of infection.

### 1.4. The Optimal Drug Release Profile for Treating Prosthetic Joint Infection

The goal of designing a drug delivery system to deliver antibiotics is to ensure that it can release enough antibiotics and eradicate the biofilm infection. This will include releasing antibiotics at a high rate that is enough to kill all the bacteria inside the biofilm without being toxic to the host. As discussed before, the ideal concentration is determined clinically by calculating the concentration needed to kill the planktonic bacteria in vitro, but this only gives the minimum inhibitory concentration (MIC) and not the minimum biofilm eradication concentration (MBEC) [25]. Unlike MIC, MBEC has not been clinically determined, is not standardized, and the time of exposure required for eradication of biofilms has not been reported. Physicians aim to achieve high concentration locally in order to eradicate the biofilm infection, so they often try to estimate the concentration and time needed for antibiotics to be effective in removing infections. The concentration and duration also depend on the antibiotics administered. In one study, it was reported that the in vitro MIC for vancomycin for treating *S. aureus* ATCC 29213/ATCC 35556 strain is 1 mg/L, while the MBEC for the same strain is >1024 mg/L, which is about 1000-fold greater than MIC. Moreover, for the same strain cefazolin, MIC is 0.5 mg/L, while its MBEC is >1024 mg/L [25]. However, these numbers do not actually reflect the in vivo and do not reflect how the duration of administration can change the concentration of antibiotics needed, as most of these tests were performed in 1 day. Muñoz-Egea et al. (2015) compared the effects of ciprofloxacin on different strains of mycobacteria in planktonic and biofilm forms [26]. They found that the MBEC for some strains was >30,000 times higher than the MIC, while for some strains, the MIC was >100 times the MBEC. This clearly shows that bacteria strains differ greatly in their susceptibility to antibiotics, and generally treatment should be offered after identifying the offending bacteria.

## 2. Diagnosis

The diagnosis of PJIs depends on several factors that start with determining the presence of an infection and its onset. Clinicians have to determine when the infection started and its severity by performing different clinical tests to look at peripheral blood, synovial fluid, the tissue surrounding the prosthetics, and other preoperative measures that will help to determine the course of treatment for the infection. The symptoms of PJIs vary depending on the onset of the infection; early infections often show symptoms of wound drainage, pain, swelling, delayed wound healing and may also manifest through fever and chills. In comparison, late-onset infections will manifest in chronic pain or implant loosening or other problems with the implant [27] PJIs are usually hard to diagnose, prompting clinicians to ask themselves whether symptoms are caused by an infection or if they are caused by something else. Once they establish that there is an infection, they have to determine the organisms causing this infection, as this is crucial in finding the most appropriate treatment for it [5]. The clinicians will often use different diagnostic tests that can confirm an infection such as erythrocyte sedimentation rate and C-reactive proteins, which are both indicators of inflammation, a sign of infection. The clinicians will also perform other peripheral blood tests that can confirm their suspicions followed by imaging technologies that can help them to visualize the infected area [5]. There does not appear to be a routinely used clinical or laboratory test that can achieve ideal sensitivity and specificity to diagnose a prosthetic joint infection [5]. The consensus among clinicians is that if two positive cultures are indicated through the tests, it is then safe to assume that there is an infection [27]. The next step for them is to find the most appropriate ways to manage the infection. 

## 3. Managing PJI

There are many challenges in managing prosthetic joint infections. Treatment options are often difficult to implement, expensive, invasive, and often cause an increase in morbidity [28]. Management of PJI should be individualized based on the patient’s case, and although there are common practices that help to manage the infection, it is highly recommended to consult a team of experts on each case to come up with the best solution for that patient [29]. The first step in managing a joint infection is the debridement, antibiotics and implant retention (DIAR) method [5]. Debridement is one of the most important steps in the management of infections. Surgeons will often perform open debridement and irrigation, washing all the implant parts and removing any parts that may be contaminated. Then, patients are placed on antibiotic suppressive therapy, as studies suggest that antibiotic treatment along with debridement can reduce the chances of implant failure [5]. Patients who have short-term symptoms and have a stable implant are often treated with DAIR. However, infections with some bacterial species can increase the chances of implant failure even with DAIR (species such as *Staphylococcus aureus* and other antimicrobial-resistant strains of bacteria such as methicillin-resistant *S. aureus* (MRSA)). If the implant is infected with species that are antimicrobial-resistant, there is a high chance of implant failure when using the DAIR method, as it has a success rate of 30–81% depending on the species, usually closer to 30% [5]. There are two main ways for the DAIR procedure application: a single-stage procedure and a two-stage procedure [29,30]. 

The single-stage procedure involves opening the joint, taking samples for histology, followed by complete removal of the implant while avoiding damaging the surrounding bone. An aggressive debridement is performed by removing all infected tissue and implants and then more cleaning followed by covering the wound with an antibiotic seal, allowing the surgeons and the staff to leave the room and disrobe [30]. This is followed by the introduction of new equipment into the operating room to proceed with implanting a replacement for the infected implant. The staff are required to scrub again to re-enter the room and continue with the procedure. The wound is reopened, new implants are implanted, and then a topical antibiotic is added, usually with an allograft such as calcium sulfate or other polymers compatible with the bone. Sometimes a polymethylmethacrylate (PMMA) is also added to support the implant. The one-stage implant exchange has many advantages including reduced cost, number of procedures, operation time, and better movement after surgery and less morbidity [30]. It is usually performed if the patients have a good remaining bone stock, if the surrounding tissue is in good condition, and if the pathogen is susceptible to antimicrobials administered orally and locally. It is also mainly used for hip implant infections and not for knee infections [5]. 

The two-stage procedure involves removal of all infected tissue and implants followed by insertion of an antibiotic-impregnated spacer during the first procedure. The patients are then given a six-week antibiotic treatment. This is followed by the second procedure involving the removal of the spacer and the addition of the new implant [27]. However, the use of spacers is debatable among clinicians, as some think that adding a spacer may worsen the outcome, especially if the infection is caused by MRSA or small colony variant, as it may increase the chance of reinfection [27]. The outcome of the surgery is largely dependent on the patient’s health, the integrity of the soft tissue, the number of previous surgeries, and other factors [31]. The surgeon usually sets a time for the second procedure once the erythrocyte sedimentation rate and the C-reactive protein rates are back to normal levels; the rates of these proteins are usually elevated, when there is an infection in the body. The two-stage procedure still has a failure rate of 28%, with a chance of reinfection.

Lichstein et al. (2014) discussed in their paper the merits of using one-stage vs two-stage procedure for managing PJIs. They indicated that two-stage procedures are often performed at a higher frequency than the one-stage procedures, especially in North America [17]. The two-stage procedure is preferred when the bacteria that cause the infection is unknown, or when the infection reaches the systemic circulation, sepsis, or when the invading organism is difficult to treat, such as MRSA. It is also used if there appears to be an infection indicated by the presence of a sinus tract, but it is difficult to determine the organisms responsible for the infection [17]. However, Wolf et al. (2011) discussed the use of a one-stage vs. two-stage procedures to treat total hip arthroplasty. They find that the use of a one-stage procedure is more effective in treating the infection than a two-stage procedure, as it increases the chances of a successful outcome and decreases mortality [32]. Both procedures have a comparable success rate; the first stage has a reinfection rate of 7.8%, while the second stage has a reinfection rate of 8.8% [30]. As both procedures have strengths and weaknesses, the determining factor may come down to cost, one-stage procedures are less costly and have shorter healing time, which makes them more attractive for surgeons and easier for patients who only have to recover once. Moreover, the surrounding tissue may become more damaged by multiple procedures [30]. Table 1 summarizes the differences between one-stage and two-stage revision surgeries.

## 4. Local Antibiotics Therapy

Local drug delivery consists in using a carrier that carries a therapeutic molecule to a specific area to deliver the molecule at the desired time and rate. The concept of local drug delivery has often been linked to the controlled release of therapeutic molecules in a time-dependent manner in a specific location in the body [33,34]. There are several factors that make finding the right material to design a drug delivery system a challenge, including the capacity of the material to degrade completely without releasing harmful substances [33]. Another factor is the capacity of the material to load the required amount of drugs to be effective in treating the disease, and it has to be biocompatible [34]. The carrier should be capable of delivering enough drugs to do the desired effect without delivering it too quickly, as the therapeutic agent will become toxic. There are many advantages for local delivery of drugs including avoiding inactivation by circulating enzymes, excretion by the kidney, leading to systemic toxicity, minimum loss of drug therapeutic activity, and improving the therapeutic outcome by directly targeting bacteria. The drugs can be added during the fabrication of the composite, so that they are entrapped inside the carrier, or they may form hydrogen bonds that will break as the composite degrades in the body [34].

Research is focused on developing drug delivery carriers that can load antibiotics and deliver them over an extended period of time. The purpose of those carriers is to deliver the drugs, biodegrade, and maintain release for a long period. Carriers that have antibacterial properties themselves can help to eliminate biofilms more effectively by working synergistically with the antibiotics to treat the infection. Figure 2 shows drug release kinetics that are involved in delivering antibiotics locally to fight infection. Drug release depends on several factors that include surface area, drug diffusion rate [35], carrier mode, and rate of degradation, as well as swelling and creation of the pores [36]. Some carriers can release drugs based mainly on a combination of factors, as represented in Figure 2. First, the (A) shows diffusion-based drug delivery diagram shows the drug is released in a short period of time as the material diffuses out of the material before it degrades. This method usually occurs when hydrophobic or small hydrophilic or hydrophobic antibiotics are added. These antibiotics will be released quickly in a short period of time. The second diagram (B) represents bulk erosion, which means that the degradation will occur relatively fast as pores are created, and there will be a late onset of large burst release. The third diagram (C) represents a surface eroding carrier that releases antibiotics as it degrades layer by layer [36], slowly and steadily releasing the antibiotics over the desired period. 

One of the ways that prosthetic joint infections are managed is by systemically delivering the antibiotics as well as locally. Local application of antibiotic achieves local antibiotics concentrations higher than the minimum inhibitory concentration (MIC) of the pathogens. Antibiotic activity against these pathogens can be time-dependent or concentration-dependent. Time-dependent antibiotics need an extended period of delivery to effectively remove the bacteria, examples include beta-lactams and macrolides, while concentration-dependent antibiotics need to achieve a specific, high concentration to effectively remove bacterial infections. Concentration-dependent antibiotics include vancomycin and aminoglycosides [28]. 

### 4.1. Bone Cements

The golden standard for delivering antibiotics to treat PJIs is by using a bone cement usually polymethyl methacrylate cement (PMMA). PMMA has been used for 35 years as an antibiotic carrier, before that it was used in the 1870s to fix a total knee prosthesis [28,38,39]. Later, it started to be used to deliver antibiotics locally at the site of infection. PMMA is usually prepared by mixing a polymer powder with a monomer liquid, resulting in an exothermic reaction that produces a rigid material [28]. When PMMA is used to deliver antibiotics the polymer powder is mixed with antibiotics powder before the monomer liquid is added to incorporate into the material before it hardens [38]. Entrapped antibiotics are released in two phases of diffusion. The first phase is usually called burst release, which is when all antibiotics near the surface of PMMA are released within minutes or hours of implantation. The next phase is when antibiotics that are trapped deeper in the material lead to a sustained release for a few days after water molecules enter the polymer and release any water-soluble antibiotics inside it. The difference between the two phases is that the first causes most of the antibiotics to be released at once, while the second releases significantly lower concentrations of antibiotics for a longer period [28,40,41].

There are only a limited number of antibiotics that can be loaded into PMMA. These antibiotics have to be heat stable because of the exothermic polymerization of PMMA [40]. Antibiotics must also be water soluble so that water molecules can dissolve them when cement is used [39,41]. Other factors that influence the choice of antibiotics include antibiotics that are effective against most bacteria and have low percentage of resistant species. The antibiotics also have to be non-allergenic meaning that it does not elicit an allergic response when used to treat infections in the body. Therefore, the antibiotics used are gentamicin, vancomycin, tobramycin, or erythromycin [41]. 

The properties of PMMA can be affected by the type of antibiotics added; some antibiotics will cause delays in polymerization time, mechanical strength, and stiffness, and even the concentration of loaded antibiotics can be affected by the type used [28]. Depending on the type of infections, surgeons might decide to use one or two antibiotics that are bactericidal and broad-spectrum. PMMA-loaded cement is usually added to the infected area in the form of beads tied in a rope so that it can all be removed at once after treatment [39]. These beads have a burst release of most of the antibiotics in the first couple of hours after implantation, releasing a concentration of 300–400 µg/mL depending on the number of beads implanted. The number of beads usually depends on the space around the infected implant. The beads have to be removed after several weeks because they will release sub-MIC concentrations of antibiotics, which can lead to antibiotic resistance of the remaining pathogens that can make future infections more severe and deadly [28]. 

### 4.2. Degradable Antibiotics Carriers 

Degradable antibiotic carriers are an attractive replacement for PMMA, as they will not need to be removed and can release all the loaded antibiotics as they degrade. One of these carriers that has been used for decades is calcium sulfate hemihydrate (CaSO_4_·1/2 H_2_O). Calcium sulfate is biodegradable, can release all of the loaded antibiotics and load a variety of antibiotics unlike PMMA [38,39,40,41,42]. The type of calcium sulfate used to deliver antibiotics is the hemihydrate in its alpha variant, as studies show that this type is preferable to the beta variant, because it is less porous and has a more predictable resorption behavior [41]. It is usually made into pellets that have a high surface area to allow for a more consistent release of antibiotics. Porosity of the material is an important factor in determining its capacity to release the antibiotics. Antibiotics can diffuse from the pellets, as there do not appear to be any chemical bonds between the antibiotics and calcium sulfate. Since there are no chemical bonds between antibiotics and calcium sulfate they are usually free to diffuse in a burst release [41]. Most of the antibiotics loaded into the material will be released within the first few hours of implantation, while the remaining will be released as the material is degrading. 

There are other biodegradable materials that are being examined for local delivery of antibiotics. Table 2 lists a few of the materials that can be used for the treatment of prosthetic joint infections including collagen fleece which has been tested as a carrier, and it is produced from sterile animal skin or from the Achilles tendon. It is biodegradable, biocompatible, and non-toxic [43]. Collagen fleece, however, has some stability problems and usually releases almost all of the loaded antibiotics in less than 2 h, which means that it would not last long enough to eradicate the bacterial infections. Some other options include synthetic polymers such as poly L-lactic acid (PLLA) or poly lactic -glycolic acid (PLGA), which are broken down into monomers and then metabolized into oxalic acid or carboxylic acid and water. The breakdown of these polymers allows for the drugs to be released as they are metabolized [44]. However, the acidic by-products produced can be toxic and can lead to inflammatory response in the host. PLGA degradation leads to the breakdown of the polymer into its building blocks of lactic acid, and glycolic acid, which leads to a reduction in the pH of the region. A lower pH leads to further degradation of the polymer and the release of the loaded antibiotics at a high rate. The polymer will also affect the functional efficiency of the antibiotics [44]. 

### 4.3. Challenges Faced by Current PJI Treatment Methods

PMMA has been the standard treatment of PJIs for a long time. PMMA bone cements are reliable and can deliver antibiotics locally to the infected area. However, it has many limitations that make it undesirable for use. First, it is not biodegradable, meaning that PMMA will not dissolve inside the body and will need to be removed after it delivers the antibiotics [60]. Second, PMMA does not release all the antibiotics trapped inside, decreasing its therapeutic potential; it releases antibiotics within the therapeutic range only for 24–48 h after implantation, which will lead to the increase of more antibiotic-resistant bacteria that are harder to treat [60]. Third, PMMA can lead to fibrin formation after implantation, which becomes a nidus for infection; the beads will colonize bacteria and can cause an infection when implanted for too long. Because PMMA is not biodegradable, it will release the antibiotics trapped inside within the first few days, afterwards it will become another foreign object where bacteria can develop biofilms causing more problems for the patients. Forth, the beads leave dead space when they are removed [47]. Furthermore, biofilms on the surface of PMMA will lead to joint dislocation, spacer fracture, and fracture of surrounding bone. Lastly, PMMA has also been associated with postoperative renal injury [60]. Yang et al. (2019) reported that PMMA spacers lead to “joint instability, increased energy demand during gait, soft-tissue contractures, and abnormal stress on the spine from pseudoarthrosis” [60,61]. 

Another bone cement that is used in the treatment of prosthetic joint infections is made of calcium sulfate. Calcium sulfate is biodegradable and bioresorbable, which makes it easier to use as it dissolves after implantation. However, it experiences many problems, including initial burst release of antibiotics, releasing most of the loaded antibiotics within the first days after implantation. It also leads to wound drainage, which may need surgical intervention. Kallala et al. (2018) hypothesized that when CaSO_4_ is placed into the infected area it alters the osmolarity of the area leading water to move out of the surrounding cells and to accumulate in the wound area leading to wound drainage [41]. They also reported on cases of hypercalcemia after the addition of CaSO_4_ beads. Several patients suffered a transient state of hypercalcemia postoperatively that needed to be treated with bisphosphonate saline solution [61]. In addition, the use of CaSO_4_ has been reported to lead to heterotopic ossification, which is an abnormal growth of bone that is usually fixed in the second-stage exchange arthroplasty [62]. CaSO_4_′s osteoconductive properties should be considered limited, as its capacity to form new bone has not been clearly demonstrated. The degradation products are also considered mildly cytotoxic due to the persistent wound drainage discussed [9]. Tarity et al. (2022) investigated the outcomes of performing the DAIR method with or without calcium sulfate beads and found that using these beads does not improve on the DAIR outcomes. They also evaluated the cost effectiveness of using the beads and found that they are not cost-effective [63], which makes them less than ideal for local delivery of antibiotics.

Most of the available degradable antibiotics carriers often suffer from many problems that make them unsuitable for treatment of PJIs. These problems include the initial burst release of antibiotics followed by a release of sub-inhibitory concentrations of antibiotics; some carriers degrade completely and release all their contents within the first day of being implanted, some produce toxic by-products along with other complications. Surgeons have been looking for a more sustainable material that can deliver antibiotics in a timely manner, is affordable and can help bone to regrow after surgery.

## 5. Bioactive Glass for Drug Delivery Applications

Treatment of prosthetic joint infection requires high concentrations of antibiotics over a prolonged period to eradicate bacterial biofilm infections. The use of a biodegradable material that can deliver high concentrations of antibiotics is essential. Surgeons prefer to use one product that can locally deliver antibiotics and degrade without having to worry about leaving nonbiodegradable material in the joint that causes future problems. Bioactive glass is a type of biomaterial that was first synthesized by Hench in 1969 [64]. A bioactive material is any material designed to cause a specific biological activity. It is the type of material that undergoes a surface reaction that will lead to the formation of a hydroxyapatite-like layer that can form a bond between soft and hard tissue [53]. The bioactivity of the materials can be tested in vitro by immersing them in stimulated body fluid (SBF). If it produces a hydroxyapatite-like layer, then the material is considered bioactive. Bioactivity in vitro is a good indication that the material will be bioactive in vivo [53]. Hench discovered that bioactive glass could form bonds and integrate with bone, which makes it suitable for bone grafts and for correcting bone defects [53]. Bioactive glass or “bioglass” is a synthetic silica-based biocompatible material with great bone bonding capacity and mechanical properties. The first glass composition made by Hench was the 45S5 Bioglass with the composition of 45SiO_2_–24.5Na_2_O–24.5CaO–6P_2_O_5_ (wt%), which is what Hench and colleges settled on as the ideal composition of bioglass [64]. This composition has high silicon content as well as the relatively high ratio of CaO/P_2_O_5_ which makes the surface highly reactive in a physiological environment [57]. 

Bioglass usually consists of varying concentrations of Na_2_O-CaO-P_2_O_5_-SiO_2_. Silicon is the main component and is usually used at the highest percentage somewhere between 40–50% of the bioglass composition and even higher for the chemically synthesized glass. There are two main ways of making the bioactive glass: melt-quench and sol–gel. When Hench first discovered bioactive glass, he made it using the melt-quench method, which requires subjecting raw material (different types of oxides) to really high temperatures around 1500 °C, the high temperatures lead to the decomposition of the oxides. The melt is then quenched by adding it to cool water to create glass frit, which is left to dry and then crushed to make the bioglass particles [64,65,66]. The sol–gel method or chemistry-based synthesis involves adding metal oxide precursors to an acidic or basic solution to induce the hydrolysis and condensation reactions of the mixture, which results in the formation of a solution or a “sol”. This is followed by the gelation of the resulting solution at room temperature until a clear gel is formed, followed by the aging of the material to avoid cracks in it [66]. The next step is to dry the glass and remove any impurities and the solvent used to obtain the bioactive glass by calcination [67]. Melt-derived bioactive glass with more silicon at 45–58% will guarantee bonding to bone and soft tissue, but any higher percentage of silicon will lead to bonding to only bone or even become biologically inert [52]. Sol–gel-derived bioactive glasses can have higher percentages of silicon without hindering their bioactivity [67]. Table 3 summarizes the differences between melt-quench and sol–gel methods by comparing the temperatures required, surface area of the produced bioactive glass, bioactivity, and some of the limitations of each method. 

Bioglass has angiogenic and osteogenic potential that is needed to reconstruct bone defects. When the bioglass interacts with biological fluids it results in the release of granules from its surface leading to an increase in osmotic pressure and pH, thus creating a hostile environment for microbes without affecting host tissue. Adding therapeutic ions such as copper (Cu^+^) and zinc (Zn) to the glass will lead to antibacterial properties that can help to combat infections. Bioactive glass used along with antimicrobial therapy seems to be a promising method for treatment of PJIs or chronic bone infections [48,66,67,68,69,70,71]. 

Bioactive glass is considered biocompatible because it is not rejected by the body when implanted. Biocompatibility is defined as “the ability of a material to perform with appropriate host response” [72]. Bioactive glass can integrate with the bone in the body and stimulate bone growth, which makes it biocompatible and bioactive [72]. According to Hench (2014), the biocompatibility of bioglass has been demonstrated though in vitro and in vivo studies which have been approved by the FDA [73]. When bioactive glass is exposed to physiological fluids, it develops a bone-like apatite layer that is similar to the mineral phase of bone, usually starting within the first 12–24 h after its inserted into the body [74], as shown in Figure 3. When bioglass is placed inside the body, it interacts with biological fluids, leading to the exchange of network modifier ions (Na^+^ or Ca^+2^) from bioglass with ions from the body fluids such as H^+^ or H_3_O^+^ [75,76]. The exchange of these ions will lead to changes in the pH level in the microenvironment to go from 7 to 10. This change in pH results in alkaline surroundings, as well as enhancing the osmotic pressure because of the change in the salt concentration. This chain of actions improves the antibacterial properties inhibiting the growth on implants. The increase in local proton concentration leads to the formation of silanol (Si-OH) groups resulting from the hydrolysis of the silica. The condensation and re-polymerization of the Si-OH groups at the surface of the granules lead to the development of a silica–gel layer [48]. This silica–gel layer will thicken as a result of the exchange of alkali ions, and it will have a negative charge that will attract the calcium and phosphate from the body fluids, creating a layer of rich amorphous calcium phosphate on top of the silica gel. The calcium phosphate phase binds to surrounding hydroxide and carbonate anions, leading to its crystallization [48,73]. Then it gradually transforms into a carbonated hydroxyapatite (HCA) layer, which resembles natural bone. Moreover, the bioactive glass plays an important role in regeneration of bone by recruiting osteoprogenitors, which will differentiate into osteoblasts driven by osteo-stimulating properties of the bioglass, this will lead to an increase in local bone healing and remodeling [52]. The osteostimulatory properties of the bioglass refer to its capacity to recruit and stimulate osteoblasts to form bone, while its osteo-inductive properties refer to its capacity to recruit stem cells and differentiate them into osteoblasts [74,75,76].

Some studies discuss the effectiveness of the use of bioactive glass in the treatment of osteomyelitis. Romano et al. looked at the effectiveness of using bioactive glass BAG S53P4 in treatment of osteomyelitis, as this type of glass has been approved in Europe for use as a medical device [74]. They used bioactive glass and no local antibiotic treatment, comparing it to the use of two calcium-based antibiotics delivery systems, and found that the bioglass treatment group showed similar effectiveness in treating osteomyelitis. Their study shows the effectiveness of bioactive glass in the treatment of bone infections [74]. Another study by Lindfors et al. (2010) was conducted on bioactive glass (BAG-S53P4) to evaluate its effectiveness as a bone graft substitute to treat osteomyelitis. The team experimentally determined the effectiveness of this bioglass by using the bioglass to treat six patients with chronic infections diagnosed with osteomyelitis; the infected bone was removed and filled with BAG-S53P4. They concluded at the end of their study that the bioglass used could successfully fill the space affected by the chronic infection, and that the study showed promising results for future use in similar applications [77]. Bioactive glass has also shown some success in treatment of periodontal osseous defects, reconstruction of cranial and maxilla-facial defects, and in creating middle ear prostheses [78,79]. 

### Review of the Literature on Bioactive Glass as Drug Delivery System

Bioactive glass has been used in various applications to repair bone and periodontal defects, as well as lesions in the skeletal system [55]. Bioactive glasses provide a new, cost-effective, and advantageous biomaterial to be used in drug delivery. Some of the advantages of using bioactive glass include that it has different compositions, different dissolution rates, and can be shaped in different forms, such as powder, films, and pellets [80]. Bioactive glass can deliver drugs to areas that are poorly innervated by blood vessels, which increases the effectiveness of treatment. Bone cancers and infections are among the main diseases in which researchers are looking at bioactive glass as a drug delivery biomaterial because of its superior bone-bonding capacity, and the possibility to control its degradation and release of loaded drugs [81]. There is evidence that bioactive glasses have a superior capacity to regenerate osseous tissue following damage to dentures caused by periodontal diseases or even after extraction of teeth [82]. 

Domingues et al. (2004) investigated the delivery of tetracycline, which is a broad-spectrum antibiotic, using sol–gel bioactive glass [55]. They also looked at the delivery of tetracycline in cyclodextrin. A sample of bioactive glass was made using the sol–gel method, where metal precursors tetraethyl orthosilicate (TEOS), triethyl phosphate (TEP), and calcium chloride with a molar ratio of 80:4:16, respectively, were used to create the bioactive glass. The drug tetracycline was added to this mixture at the end to load the material with the drug. This was followed by testing the release rate of the bioactive glass by testing the bioactive glass in simulated body fluid (SBF) for 80 days. Bioactive glass was also tested in vivo using female C57BL/6 mice. Both in vitro and in vivo experiments were performed in triplicates, and the control included bioactive glass that was not loaded with tetracycline. The loaded glasses showed a burst release of about 12% in the first 8 h followed by a steady slow release of a cumulative 22% for tetracycline and 25% for tetracycline in cyclodextrin, which means that the bioactive glass can deliver drugs effectively for 3 months. In vivo studies show that the bioactive glass loaded with the drug did not have a significant inflammatory effect that would be of concern for its use as a drug delivery system [55]. 

Geurts et al. (2019) looked at the costs of using one-stage vs. two-stage treatment of chronic osteomyelitis. They compared bioactive glass, specifically Bonalive^®^, with PMMA spacers, where the bioactive glass is used in the one-stage procedure, and the spacers are used in the two-stage procedure. Bonalive^®^ is a melt-quench type of bioactive glass that has a composition of 22.7 Na_2_O, 21.8 CaO, 1.7 P_2_O_5_, 53.9 SiO_2_. It has osteoconductive properties, can form new bone, and antibacterial properties. The antibacterial property of the bioactive glass comes from the local increase in pH caused by the dissolution of the bioactive glass granules. The increase in local pH creates a hostile environment for bacterial adhesion and proliferation [83]. Their study was focused on the costs associated with one-stage vs. two-stage osteomyelitis, as finding an effective treatment should also consider the costs associated with the treatment used. They found that the costs associated with PJI decreased when bioactive glass was used because of “decrease in hospital stay, less surgeries involved, and lower antibiotics cost” [83]. Bioactive glass not only offers a new treatment option for osteomyelitis, but it can significantly reduce costs on the healthcare system and the patients. In a similar study, Romano et al. (2014) found that treatment involving bioactive glass without antibiotics, Bonalive^®^, showed similar biofilm eradication rates and less drainage, compared to calcium-based bone substitutes [74]. 

Soundrapandian et al. (2010) investigated a bioactive glass scaffold for local drug delivery for the treatment of osteomyelitis and infection of the bone [81]. Treatment requires the delivery of antibiotics over a prolonged period, 4–6 weeks, to eradicate the infection. The group developed bioactive glass using the melt-quench method with a composition of 45% SiO_2_, 21.2% CaO, 26% Na_2_O, and 7.8% P_2_O_5_. The scaffold was made by mixing the glass powder with porgen and compressing it and finally cutting it into circular disks. The scaffold was then coated with chitosan which is meant to regulate the release rate of the loaded drug from the scaffold. The scaffolds were also loaded with gatifloxacin and fluconazole by the vacuum infiltration method where the scaffold is immersed in a solution containing the drugs then vacuum is applied for some time to achieve the highest loading possible. They tested the capacity of scaffolds with different heights to release the drugs and found that the scaffold with increasing height, the drug release is decreased, which is explained by the fact that the drug will have to travel a longer distance to exit the scaffold so it will take more time to be released [81]. The drugs are likely to be present inside the scaffold in four states: attached to the exterior, at the pore opening inside the scaffold, attached to the pore wall, and deeper inside, because of the pressure that applied when the scaffold was being synthesized. They also found that coating the scaffold with chitosan reduced the initial burst release of the drug significantly, and at 0.5% chitosan coating, the scaffold reduced the initial release of the drug as well as extend the release of the drugs above minimum inhibitory concentration for the entirety of the study [81]. 

Nandi et al. (2009) looked at the release of cefuroxime axetil from the bioactive glass [82]. The group designed a bioactive glass composite and loaded it with cefuroxime axetil by vacuum infiltration and tested the release rate of the composite in vitro and in vivo using rabbits with the infected tibia to induce osteomyelitis. The bioactive glass was prepared by the melt-quench method, followed by mixing them with naphthalene to make the composite, which was pressed using a cold-isostatic press. The naphthalene was then removed by heating the composite followed by using the vacuum infiltration method to mix in the drug. The drug in vitro release was determined over 21 days; however, it was observed that the drug eluted within 6 days with most of it released within the first 3 days. In vitro studies were performed in phosphate buffer rather than simulated body fluids, which may explain the quick release of the drug, compared to other studies. The in vivo studies showed that new bone was formed in the experimental rabbits and that the release showed the highest value on day 21 and continued by day 42. The release rate was measured by looking at the level of the drug in the bone and the serum, which are both promising results for bioactive glass, as the release continued above the minimum inhibitory concentration for the in vivo study even on day 42 where the release was significantly lower than the release on day 21, the highest release [82]. 

Araújo et al. (2017) examined the capacity of melanin-coated bioactive glass scaffolds to load ibuprofen and its mechanical strength in vitro [83]. In their study they were looking to develop a scaffold that could act as a template for bone regeneration and development. To provide structural support for bone growth, the scaffold needs to be macroporous, have mechanical strength, be osteoconductive, and osteoproductive. They looked at the fabrication of a bioactive glass scaffold coated with melanin and loaded with ibuprofen using supercritical CO_2_ technology [83]. The bioactive glass used in this study was made using the melt quench method starting with Ca_3_(PO_4_)_2_, Na_2_CO_3_, KNO_3_, CaCO_3,_ MgO, and CaF_2_ powders. The scaffolds were made using foam replication methods using a polyurethane template. The release kinetics of the melanin-coated bioactive glass and the uncoated scaffold were compared to show the effect of melanin coating on the release of the drug. The melanin-coated scaffold showed a controlled drug release, compared to the uncoated sample, as shown in Figure 4. 

## 6. Mesoporous Bioactive Glass

Following the success of bioactive glass and its versatility, Yan et al. (2004) investigated developing highly ordered mesoporous bioactive glass (MBG), which has a high pore volume, and surface area [84]. MBG has many advantages over other bioactive glasses including being made at lower temperatures, improved homogeneity, wider range of bioactive compositions, and adjustable mesoporosity, and purity. Mesoporous bioactive glass bioactivity is determined by its compositions. Different compositions dissolve at different rates to form a hydroxyapatite layer [42,84,85]. MBGs are more suitable for drug delivery because of their ordered mesoporous structure, large surface area, and great loading capacity. The ordered mesoporous structure ranges from 2 to 50 nm, creating channels that can load drugs and allow for their controlled release. The mesoporous structure uses structure-directing agents to create the mesopore structure [57,85].

Mesoporous bioactive glass has a similar composition to that of Hench’s bioglass but has a higher bioactivity because of its “ordered structure, large specific surface area, and high pore volume” [42]. MBG is prepared by the sol–gel method with the addition of surfactants to create the mesopores. To create highly ordered structures, surface-directing agents are added during the sol–gel process. These surface-directing agents include cetyltrimethylammonium ammonium bromide (CTAB), poly (ethylene glycol)-poly (propylene glycol)-poly (ethylene glycol) (P123), and F127, which are used to make the bioglass. The order of the material depends on the chemistry of the agent (ionic, non-ionic) [86], its concentration [87], temperature of the solution, and pH, as the well as organic: inorganic volume ratio of the materials used to make the solution [68]. The agents self-organize their structure to create micelles that often contain hydrophobic inner regions and hydrophilic outer regions, as seen in Figure 5. The hydrophilic regions link hydrolyzed silica and self-assemble it to form ordered mesopores. The mixture of the directing agents and bioactive glass undergoes the evaporation-induced self-assembly (EISA) process. EISA leads to the spontaneous assembly of material through noncovalent bonding without intervention from external factors. Once evaporation is completed and surfactants are removed, the resulting mesoporous glass is obtained [87]. 

The hydroxyapatite layer can form in a shorter time when using MBG and has better capacity to bond to bone. Similarly, to its predecessor, MBG releases Ca and Si which promote osteoblast proliferation and differentiation. The preparation of MBG influences its drug delivery capacity, as each surface-directing agent changes the pore volume and loading capacity of the material [86,89]. In addition, the composition of MBG influences its capacity to load drugs and deliver them. Zhao et al. (2008) found that increasing the CaO content in MBG enhances the loading capacity and modulates the release of loaded drugs [90]. The degradation of the bioactive glass depends on the compositions, as glasses with higher CaO content degrade and convert to HCA faster than glasses higher in SiO_2_. The researchers published a schematic of how their drug of choice adsorbed in MBG molecules and how its interactions with calcium influenced the release rate of the drug [90]. They suggest that the tetracycline (TC) molecules are adsorbed in MBG in two ways: by physical adsorption and by chemical adsorption of TC molecules, as shown in Figure 6. The chemically adsorbed TC molecules are thought to be adsorbed through chelation with calcium ions on the pores of MBG. The schematic also predicts that physically adsorbed TC molecules are responsible for the initial release of the drug, which is faster, as it is only physically adsorbed. Chemically adsorbed molecules are harder to remove and therefore slow the release of the drug over a longer period [91].

MBG can load different therapeutic drugs via the immersion technique, which involves the immersion of the MBG particles into a solution containing a pre-determined concentration of the drug, which will lead the drugs to be incorporated into the mesopores. The drugs can also be added during the fabrication of the MBG so that it is entrapped within the structure of the bioglass, which is more efficient but more difficult to achieve [91]. The capacity of the bioglass to load drugs directly correlates to the pore size and the specific surface area of the bioglass used [90]. Bigger pore sizes will ensure that the drug particles are hosted within the mesopores and not hanging on the outside of the bioglass. The drugs are released based on a diffusion mechanism that allows the drugs to diffuse in four different states, according to Xia et al. (2006) and as shown in Figure 7. The first and fourth states indicate adsorption of the drug at the surface of the bioglass; this leads to the release of the drug quickly. This is followed by diffusion-based release of the drug (gentamicin), which is a slower release caused by the interactions of the Si-OH on the surface with the drug leading to slower more steady release rate. Hydrogen bonds between the glass and the drug will slow the release rate, and the drug will be released once the silicon layer dissolve turning into hydroxyapatite [92]. 

MBG loading efficiency depends on the composition and the formulation. Wu et al. evaluated different compositions of MBG and found that MBG particles with composition 58Si36Ca6P had a loading efficiency of 36–48% with an initial burst release of about 28–60% on the first day and could deliver gentamicin for about 10 days [89]. At the same time, 80Si15Ca5P MBG particles had a loading efficiency of 35% with a burst release of 25–45% and lasted about the same as the 58Si. MBG spheres had the longest release rate at around 14 days, but with a lower loading efficiency and a higher burst release depending on the loaded drug [89]. The composition of MBG plays a significant role in its capacity to load drugs and release it at a slower rate. Li et al. (2013) compared the loading efficiency of MBG, compared to non-mesoporous bioactive glass and found that the loading of efficiency of MBG is about 80%, while that of the non-mesoporous one had a loading efficiency of less than 20%, as shown in Figure 8. They also looked at the in vitro release of the drug and found that the MBG released the drug slowly and sustained manner for the duration of the experiment, 6 days, which remained around 70.4 µg/mL, which would be effective against the bacterial infection while non-mesoporous released 2–3 µg/mL [93].

Wang et al. (2018) examined the capacity of amino-modified mesoporous bioactive glass (N-MBG) to deliver Alendronate (AL), which is a commonly used drug for osteoporosis [94]. This drug has low bioavailability and low efficiency when delivered orally so their study was aiming to find a drug delivery system that can locally deliver the drug. In their study the researchers used an MBG scaffold but modified it post-production with an amino group that is hypothesized to give it better bone-bonding capacity and decrease the degradation rate [94]. To make the scaffold, they used polyvinylpyrrolidone (PVP), polyethylene glycol (PEG), and MBG powder. The three materials were mixed and added to a mold to shape the scaffold and pressed under 5 MPa to form the final scaffold; this was followed by adding the amino functional group by adding aminopropyltrimethoxysilane. The drug was loaded into the scaffold by immersion in a solution containing the drug. The group compared the loading capacity and release rate of both the scaffold and the amino functionalized scaffold and found that the amino-functionalized group had better loading capacity and better performance overall, as shown in Figure 9.

Using mesoporous bioactive glass to treat PJIs seems to be promising. PJIs are caused by biofilms and are one thousand times harder to treat than planktonic bacteria; therefore, they need to be exposed to high concentrations of antibiotics above the minimum inhibitory concentration for biofilms (MIC). MBGs can load a large volume of antibiotics and deliver them as they dissolve, allowing the antibiotics to be released long enough to treat infections. MBG has antimicrobial abilities itself that can be enhanced by incorporating different metal ions such as zinc or magnesium into its composition [95]. Moreover, MBGs can dissolve and bond to bone, which is another advantage in using it for the treatment of the infection [92]. Surgeons prefer to use biocompatible and biodegradable material that can deliver antibiotics and does not need to be removed from the injury site. Most of the currently used treatments suffer from burst release and bio-incompatibility, which makes their uses limited and undesired. MBGs can offer a new treatment that will treat PJIs more effectively and will enhance their capacity to form new bones. 

Before MBG can be used to deliver antibiotics to treat prosthetic joint infection, it needs to be fully investigated in vivo. The degradability of any material is significantly different in vivo, and therefore, the dissolution rate and capacity to deliver antibiotics needs to be investigated both in vitro and in vivo. Promising results have been reported from studies that have investigated the in vivo performance of MBG. For example, Wang et al. (2019) synthesized hollow mesoporous bioglass nanoparticles and tested their capacity to deliver ibuprofen and promote bone regeneration in vivo. They concluded that the ibuprofen-loaded bioglass nanoparticles showed great bone tissue regeneration capacity in rats and could decrease local inflammation [96]. Similarly, Berkmann et al. (2020) investigated the use of MBG-loaded with bone morphogenic protein-2 (BMP-2) in vivo, which showed better bone regeneration than untreated control [97]. These studies demonstrate the potential of MBG in vivo applications, but further research is needed to investigate to fully understand its performance and limitations as a delivery system for antibiotics. 

Although mesoporous bioactive glass shows great potential in replacing current treatments for PJIs, it has some disadvantages that need to be overcome for it to be used to deliver antibiotics. One of the disadvantages of using MBGs is that there is no standard composition that is universally agreed upon as the most effective for loading and delivering the drugs. This means that every composition has a different particle size and surface area and will behave differently when tested in vivo. Mesoporous bioactive glass has not been clinically used to treat either osteomyelitis or implant infections but should be effective at treating both because of its superior capacity to load antibiotics and to deliver them over a prolonged period. Additionally, MBG has a greater capacity to bond to bone and form a hydroxyapatite layer that can help to rebuild and heal bone.

Table 4 considers different studies that looks at the drug delivery capacity of mesoporous bioactive glass. The table outlines the different drugs that were loaded into the MBG, its loading capacity and the release kinetics of the drug. The table also shows how varying the composition of MBG will affect the loading capacity and the release of the loaded drugs. 

## 7. Conclusions

Prosthetic joint infections are a devastating result of joint arthroplasties that affect thousands of people. As the population ages, more total joint replacements will be needed to help to maximize the mobility and functionality of the ageing joints. The current treatment of these infections with antibiotics is limited by their capacity to degrade and maintain constant release to eliminate the infection completely. PMMA has been considered the standard treatment for a long time, but it has more limitations than advantages. These limitations include being nondegradable, a fast release followed by a slow subtherapeutic release, and these materials can also encourage the development of biofilms on their surface. Another treatment currently in use involves calcium sulfate, which is biodegradable and capable of loading and administering antibiotics but can lead to wound drainage, heterotopic ossification, and hypercalcemia. New treatments are being developed to address these limitations. One of these new treatments involves bioactive glass, which is a synthetic bioactive silica-based material that can bond to bones and degrade completely when used inside the body. Unlike other treatments suggested, bioglass has great bone-bonding capacity and mechanical properties. A new variety of bioactive glass made using the sol–gel method and agent-directing surfactants is called mesoporous bioactive glass (MBG), which has high pore volume and enhanced surface area. MBG is considered as a drug delivery system that can replace currently used treatments of PJI. MBG can load more drugs and can deliver them to treat infections. It is biocompatible, biodegradable, have high pore volume, large surface area and great loading capacity.

## Figures and Tables

**Figure 1 pharmaceutics-15-01401-f001:**
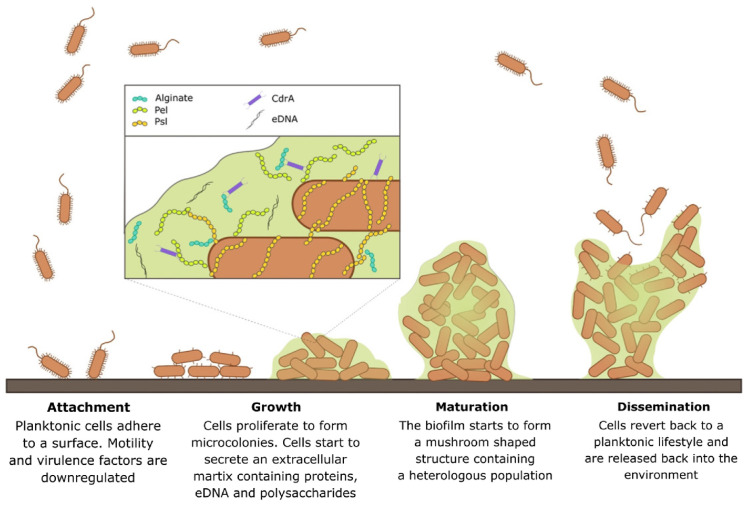
Biofilm development including cycles of growth, attachment, maturation, and dispersal [15]. The bacteria will first attach to the implant surface and suppress mortality factors but induce adhesion factors. The growth period is when the bacteria will start dividing and recruit other bacteria to create a colony. The maturation is when the biofilm grows a matrix to protect the bacteria inside, and finally dispersal will lead to detachment of some bacteria to spread elsewhere. The matrix includes exopolysaccharides as shown in the legend including alginate, Pel and PsI,. CdrA (a protein) and extracellular DNA (eDNA) add structural strength to the matrix. Reproduced with permission from Maunders, FEMS Microbiol Lett, published by (Oxford University Press), (2017).

**Figure 2 pharmaceutics-15-01401-f002:**
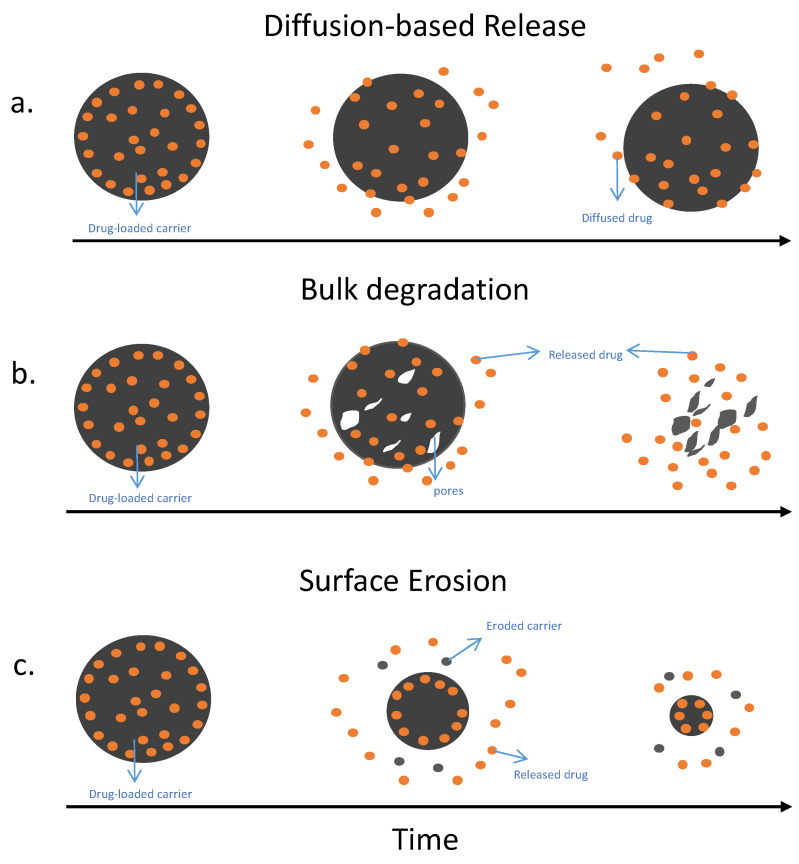
The release of antibiotics from degradable carriers can be categorized into three themes. (**a**) The first theme is diffusion-based release in which the drug is released quickly from the material. (**b**) The second theme is bulk erosion, where the antibiotics are released as the material degrades. This happens over a short period of time as the material erodes in bulk. (**c**) The third theme is surface eroding carrier materials which slowly release antibiotics as they degrade [37]. Adapted from Geraili et. al. (2021) with permission under a Creative Commons license from Wiley.

**Figure 3 pharmaceutics-15-01401-f003:**
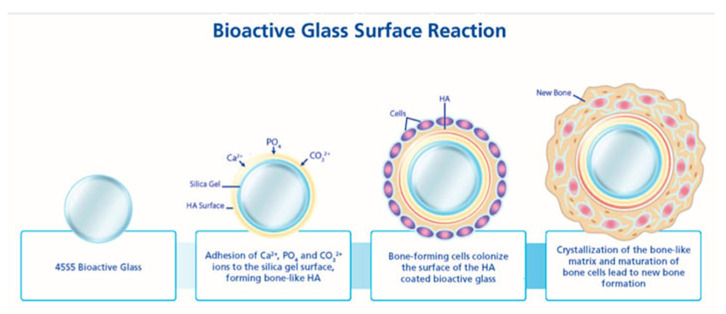
Bioactive glass reaction to exposure to physiological fluids and the formation of a hydroxyapatite layer. The reaction involves the exposure of the bioactive glass to biological fluids leading to the formation of hydroxyapatite which allows bone-forming cells to attach to the surface and promote the growth of bone to replace bone defects. Reproduced with permission Under a Creative Commons license from Wikimedia Commons.

**Figure 4 pharmaceutics-15-01401-f004:**
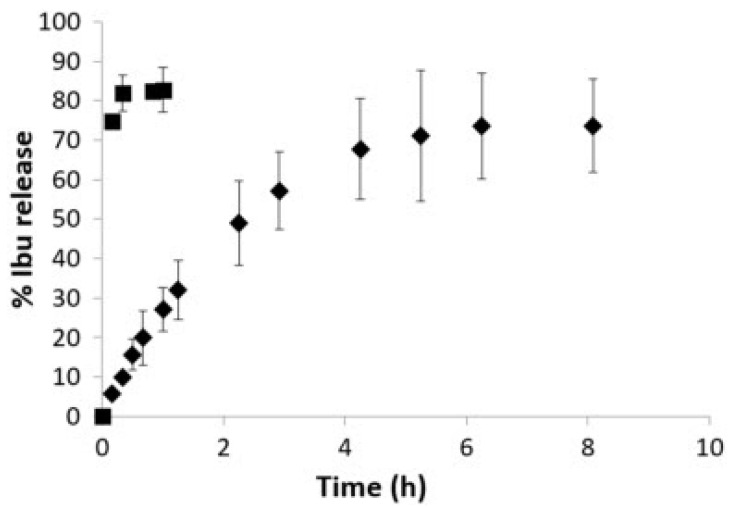
Release of ibuprofen from coated (diamond) and uncoated (square) bioactive glass scaffolds in SBF [83]. The ibuprofen release lasts for a longer period when using coated bioactive glass scaffolds, compared to uncoated. Reproduced with permission from Araújo, Materials Science and Engineering: Cl, published by Elsevier, 2017.

**Figure 5 pharmaceutics-15-01401-f005:**
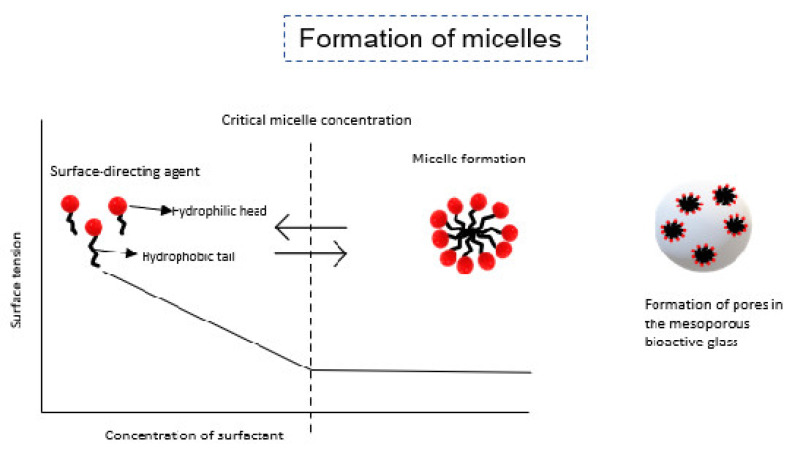
Formation of micelle as the concentration of surfactant passes the critical micelle concentration (CMC). The surfactant creates micelles with hydrophobic inner region and hydrophilic outer region. When dissolving the metal oxides, these micelles will be entrapped inside to create pores. Once the surfactants are removed, they leave behind a mesoporous structure [88]. Adapted from Hardy et. al. (2018) with permission Under a Creative Commons license from Elsevier.

**Figure 6 pharmaceutics-15-01401-f006:**
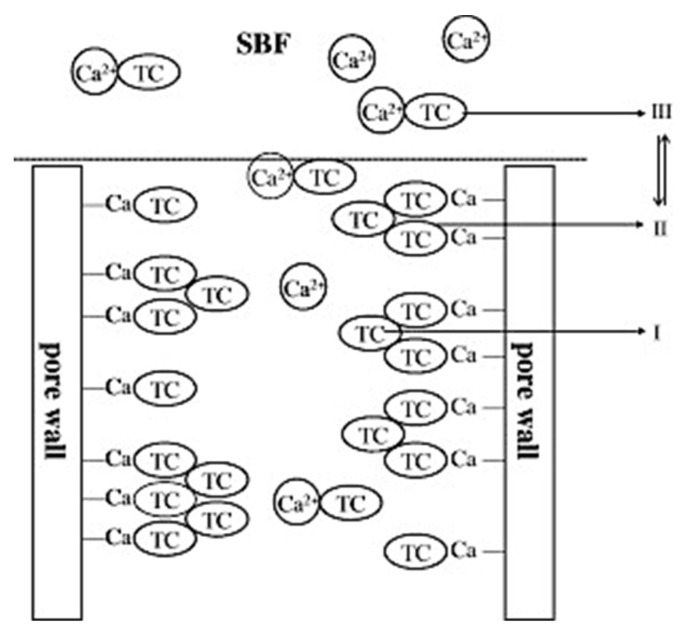
Schematic of different tetracycline (TC) molecules adsorbed in MBG molecules during drug release: (I) physically adsorbed TC particles, (II) TC chelated with Ca on the pore walls, (III) TC chelated with Ca in SBF [90]. Reproduced with permission from Zhao, Microporous and Mesoporous Materials, published by Elsevier, 2008.

**Figure 7 pharmaceutics-15-01401-f007:**
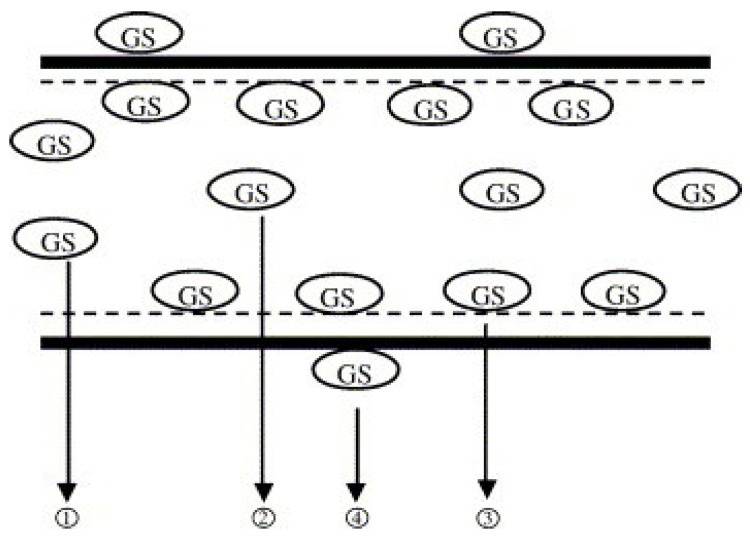
The states of gentamicin (GS) molecules hosted within MBG, including (1) GS inside the pores of the window, (2) GS entrapped in mesopores and no hydrogen bonds, (3) GS in mesopores with hydrogen bonds with Si-O and P-O, and (4) GS molecules adsorbed on external surface [92]. Reproduced with permission from Xia, Journal of Controlled Release, published by Elsevier, 2006.

**Figure 8 pharmaceutics-15-01401-f008:**
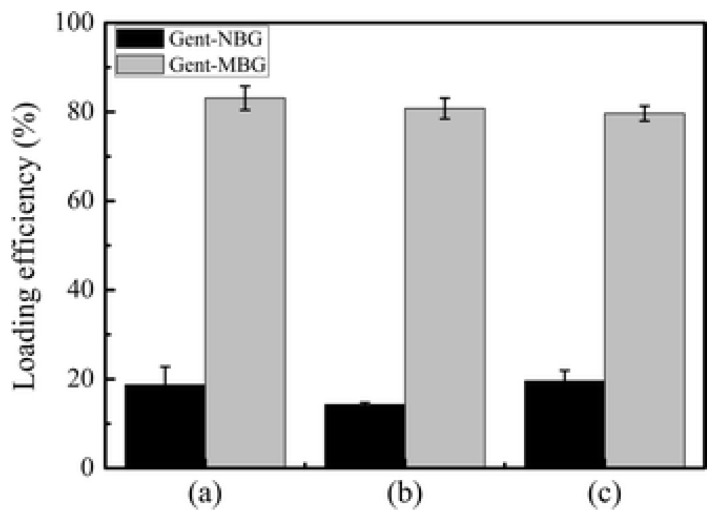
MBG drug loading efficiency compared to non-mesoporous BG in gentamicin solution with different initial concentrations [93]. The loading efficiency for the MBG is around 80%, while it is much lower for the NBG. (**a**–**c**) Different concentrations of gentamicin solutions used in the study: (**a**) 400 μg/mL, (**b**) 600 μg/mL, and (**c**) 800 μg/mL. Reproduced with permission from Li Journal of Materials Science: Materials in Medicine, published by Springer Nature, 2013.

**Figure 9 pharmaceutics-15-01401-f009:**
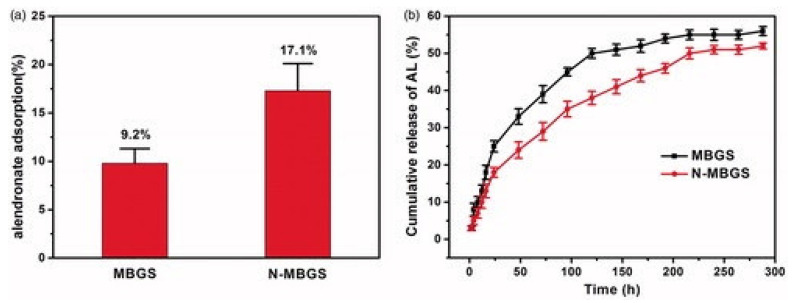
The maximum loading of Alendronate (AL) in MBGS and amino-modified MBG (N-MBGS) and the release profile of Alendronate from MBG and N-MBG. The loading capacity of N-MBG is higher than MBGS, and the cumulative release of the N-MBG is slower than MBGS [94]. (**a**) represents the maximum loading of alendronate in MBGS and N-MBGS (**b**) represents the cumulative release of Alendronate from MBGS and N-MBGS) Reproduced with permission from Wang from Artificial Cells, Nanomedicine, and Biotechnology, published by Taylor & Francis, 2018.

**Table 1 pharmaceutics-15-01401-t001:** Comparison between one-stage and two-stage revision surgeries.

	Infection Type	Success	Type of Joint	Cost
One-stage	Preferred when the infected pathogen is known	Higher eradication rates and functional outcome [30]	Hip and shoulder joints	Lower costs and shorter time at the hospital
Two-stage	Preferred when the pathogen is unknown or hard to treat	High success rate for patients with poorer bone stock	Knee joints	Higher cost related to multiple surgeries and longer hospital stay

**Table 2 pharmaceutics-15-01401-t002:** Different local antibiotics delivery vehicles and their properties and limitations.

Drug Carrier	Properties	Degradability	Elution Characteristics	Limitations
PMMA	Pre-loaded beads or by direct mixing of antibiotics into material before implantation. Can load multiple antibiotics	Non-degradable	Less than 50% of antibiotics eluted after 4 weeks [45]	Nondegradable Limited antibiotics can be used Burst release followed by subtherapeutic antibiotics release.
Calcium Sulfate	Can be 100% synthetic to remove any impurities. A wide range of antibiotics can be used. Degradable and osteoconductive. Reabsorbs within 30–60 days [46]	Degrades completely within 4–6 weeks [47]	Releases most loaded antibiotics within 72 h. Burst release 68–74% (*w*/*w*) by the second day [48]	Burst release followed by slow release of antibiotics and limited capacity to form new bones [46,49]. Can lead to wound drainage, heterotopic ossification, and hypercalcemia Hypersensitivity reaction [46]
Collagen Fleece	Biodegradable Biocompatible and non-toxic	Degrades completely within 8 weeks [43]	Burst release, releases all content within 2 h of implantation	Burst release; releases most antibiotics within 2 h of implantation Mild immunogenicity [50]
Polyesters (PLGA)	Biodegradable Highly tunable, can be tweaked to load antibiotics and deliver them efficiently by changing the ratios of the polymers	Depends on composition, can take 8 weeks to few months [44]	Burst release but maintain seroma levels above sensitivity for 55 days [51]	Can cause inflammatory response Lower pH caused by degradation can lead to really fast release of loaded antibiotics [44]
Bioactive glass	Synthetic silica-based material. Excellent mechanical and bone-bonding properties [52] and capacity to degrade at controllable rate [53] Osteostimulation and angiogenic potential Has antibacterial properties Many formulations approved for use by FDA	Depending on the composition, sol–gel glasses can take 12–52 weeks [54]	12% w burst release followed by sustained release for 3 months [55]	Limited porosity [56] Degradation and hydroxyapatite formation depend highly on compositions and production method [57]
Mesoporous bioactive glass	Use of surface-directing agents and templating methods Enhanced surface area and higher pore volume Excellent cytocompatibility [58] and better apatite formation than non-mesoporous bioactive glass	Depends on composition, has better apatite formation capacity		These have not been tested in vivo. Brittleness [59]

**Table 3 pharmaceutics-15-01401-t003:** Comparing melt quench method and sol–gel method to make bioactive glass.

	Temperature	Surface Area	Bioactivity	Limitations
Melt quench	Really high temperatures (>1500 °C)	0.15 to 2.7 m^2^/g [68]	Limited composition range of SiO_2_, biologically inert at SiO_2_ > 60%	Defects such as cracks and bubbles that may affect properties of the glass
Sol–gel	Aging 70 °C Drying 120 °C Calcination 600 °C [66]	126.5–164.7 m^2^/g [68]	Wider composition range, bioactive at SiO_2_ of 90%	Shrinkage during drying, microcracks, long hydrolysis stage Carbon traces [69]

**Table 4 pharmaceutics-15-01401-t004:** Examples of bioactive glass drug delivery systems and their loading capacity and release rate.

Materials	Drug	Composition	Loading	Release	Study
Mesoporous bioglass (MBG)	Tetracycline	Multiple by varying CaO content at 5C, 10C, 25C, and 35C	MBG 100S = 10.5% MBG 90S5C = 15.5% MBG 80S15C = 15.8% MBG 70S25C = 17.4% MBG 60S35C = 18.3%	100S—high initial release and release 50% at 1.9 h 90S5C, 80S15C, 70S25C, 60S35C equilibrium at 72 h TC total release MBG 100S = 98% MBG 90S5C = 56% MBG 80S15C = 25% MBG 70S25C = 25%, MBG 60S35C =38%	Zhao, 2007 [90]
MBG	Ibuprofen	80Si_2_O-15CaO-5P_2_O_5_	46 wt% of ibuprofens in solution	Release under static conditions 60% of ibuprofen	Zhang, 2012 [98]
Borate bioglass (sol–gel)	Vancomycin	6Na_2_O-8K_2_O-8MgO-22CaO-54B_2_O_3_-2P_2_O_5_	80 mg/g of vancomycin per gram of borate glass powder	Cumulative release of vancomycin was 85.99–94.43% and lasted for 18 days	Xie, 2009 [48]
MBG	Gentamicin	58Si_2_O-23CaO-9P_2_O_5_	35.2 wt% of total gentamicin in solution	30 wt% released after 20 days M58S had lower initial burst release in the first 24 h than normal BG (58S)	Xia, 2008 [99]
MBG With different surface modifications	Ipriflavone	85Si_2_O-10CaO-5P_2_O_5_	Phenyl 85 MBG = 11.7% OH-propyl 85 MBG = 6.08% NH_2_-propyl 85 MBG = 6.14% SH-propyl 85 MBG = 4.05% Cl-propyl 85 MBG = 1.68% Butyl 85 MBG = 1.06%	Phenyl 85 MBG = 3% OH-propyl 85 MBG = 6.02% NH_2_-propyl 85 MBG = 7.27% SH-propyl 85 MBG = 13% Cl-propyl 85 MBG = N/A Butyl 85 MBG = N/A	López-Noriea, 2010 [100]
MBG	Gentamicin		79.6–83.1% four times higher than non-MBG used as control	80% of gentamicin was released in 6 days while maintaining slow and sustained release	Li, 2013 [93]
MBG	CFS a combination of sodium ceftriaxone (CFT) and sodium sulbactam (SUL) in the ratio of 2:1.	78Si_2_O-16CaO-6P_2_O_5_	40.4% of CFS	In different pH conditions, a burst release of >50% occurred in the first 24 h followed by a slow and sustained release for the duration of the study (168 h)	Anand, 2020 [95]
Amino-functionalized MBG (N-MBG)	Gentamicin sulfate	80Si_2_O-15CaO-5P_2_O_5_	MBG = 48.9 wt% N-MBG = 62.92 wt%	MBG 7-day release 60.7% of the total loaded drug N-MBG 7-day release 45.1% with sustained release after N-MBG and MBG had initial burst release (12 h) = 30–40% of total drug.	Jiang, 2017 [101]
MBG nanospheres	Doxorubicin hydrochloride	80Si_2_O-15CaO-5P_2_O_5_	High drug loading capacity of 90%	Decreased burst release and sustained release of DOX for 180 h (duration of study) with cumulative release at 15–30% with varying pH of SBF solutions	Wu, 2013 [67]
Strontium doped bioactive glass	Ibuprofen	63Si_2_O-(37 − x)CaO-xSrO x = 0–1 mol%	High drug loading of 78.5–88.8% depending on the mol% if strontium	Burst release in the first few hours followed by steady release for 28 days	El Baakili, 2022 [102]
Nano-bioactive e glass	Ciprofloxacin	97SiO-3CaO 87SiO-13CaO 77SiO-23CaO 67SiO-33CaO	Loaded drug decreased with increasing mol% of CaO Loading efficiency of 92.5–98.63%	Cumulative release of the total loaded amount of drug for the different compositions ranges from 24.76–52.22% over 28 days	El-Kady, 2019 [103]

## Data Availability

Not applicable.

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
