# Peer review of "Prosthetic Joint Infections: Biofilm Formation, Management, and the Potential of Mesoporous Bioactive Glass as a New Treatment Option"

_pharmaceutics, 2023, doi:10.3390/pharmaceutics15051401_

Round 1

Reviewer 1 Report

The work is a thorough and well-done literature review, but I think the materials and methods part is missing here. Where the authors would describe what libraries they checked in which years, What keywords they used, How much they received in articles and how much and for what reason they used PRIZMA literature for their review.

Introduction

I like the part. It is explained here why the reaction of the body is hampered in the case of problems with implants.

Line 48

 US and the projections indicate that by 2020 it will cost it around $1.6 billion dollars

You can give exactly how much it was in 2020, because probably this data in 2023 is already available?

Line 93

Tande at al. you shad add a reference to this part it should be [5]?

Figure 1 Using abbreviations not explained under Pel, Psi?? alginate is also listed there, where does it come from? Alginate is of marine origin (algae) it is not synthesized by bacteria, please live it

Line 181

Siala at al  [20]

Line 279

DIAR methods – please explain this aberration

 Line 360

Figure 3 looks at drug release kinetics that are involved in delivering antibiotics locally to fight infection- Should be Figure 2???

Figure 2 On the Y-axis there should be some values, eg  µg/ml or similar, because the amount of released antibiotic has a unit?

Line 435 CaSO4* 1/2H2O

because it is less porous-  It is usually made into pellets that have a high surface area and high porosity to allow for a more consistent release of antibiotics- in the end, alpha has less or more pores? According to the literature, Alpha mixes with less water, so it has fewer pores. It is different if the material is prepared by the manufacturer as a ready implant, then the production process may include the phase of formation of synthetic pores, e.g. by evaporating the substance in gaseous form from the material

You can also consider PMMA-based cement, which releases ions from bioactive glasses, e.g.

Raszewski Z, Chojnacka K, Mikulewicz M. Preparation and characterization of acrylic resins with bioactive glasses. Sci Rep. 2022 Oct 5;12(1):16624. doi: 10.1038/s41598-022-20840-1. PMID: 36198737; PMCID: PMC9534886.

Line 536

The sol gel method or chemistry-based synthesis requires adding the raw material and creating a solution (“sol”) by adding  an acid catalyst to the raw material and water- This needs some clarification, we don't add Na2O or SiO2 puree to the solution and expect it to form a glass. Organic compounds are added for the teraethoxy silane precursor Si and others. There is no gel process, only hydrolysis, as a result of which particles are formed that are insoluble in the reaction medium (mixtures of water and alcohol silanes)

Figure 8 gives a, b and c but you have to explain what that means.

Suggestion. At work, use different abbreviations at the end of the manuscript, you add a table in which all these abbreviations will be explained, it makes it very easy to read. Thank you

Good luck with further research!

Author Response

MDPI Branch Office, Belgrade
Bulevar Milutina Milankovica 7v,

11070 Belgrade, Serbia

Re: Revised Manuscript pharmaceutics-2293211- special Issue “"Design of Mesoporous Materials for Biomedical Application"

Dear Editor,

We are pleased to submit our revised manuscript: “Mesoporous Bioactive Glass as a Potential Treatment Option for Prosthetic Joint Infections” by Dana Almasri and Yaser Dahman*. In the revised manuscript we carefully incorporated all changes suggested by the reviewers. In addition, we are pleased to respond point –by– point to the main comments made by the reviewers in the “Detailed Response to Reviewers” file attached.

Best regards,

Yaser Dahman, Ph.D., MBA, P.Eng

Professor, Department of Chemical Engineering

Toronto Metropolitan University

350 Victoria St. Toronto, Ontario M5B 2K3

Phone: 416-979-5000, ext. 554080

Fax: 416-979-5083

Email: ydahman@torontomu.ca

Reviewer 1

  • The work is a thorough and well-done literature review, but I think the materials and methods part is missing here. Where the authors would describe what libraries they checked in which years, what keywords they used, how much they received in articles and how much and for what reason they used PRIZMA literature for their review.

This is a review article that looks to highlight common practices used by medical professionals in the treatment of prosthetic joint infections and how bioactive glass can be used in the future. We critically reviewed more than 200 articles and highlighted information in a way that helps the reader understand all different angels of this infections and challenges of treating it. There is no experimental work that was done yet that requires “Materials and Methods” section in the present review.

  • Introduction: I like the part. It is explained here why the reaction of the body is hampered in the case of problems with implants.

Thank you!

  • Line 48: US and the projections indicate that by 2020 it will cost it around $1.6 billion dollars. You can give exactly how much it was in 2020, because probably this data in 2023 is already available?

We have implemented the recommended edit in the revised manuscript on the first paragraph of page 2.

  • Line 93 Tande at al. you shad add a reference to this part it should be [5]?

We have implemented the recommended edit in the revised manuscript.

  • Figure 1 Using abbreviations not explained under Pel, Psi?? alginate is also listed there, where does it come from? Alginate is of marine origin (algae) it is not synthesized by bacteria, please live it

We agree with this suggestion. The abbreviations have been explained under the caption (fig 1) in the revised manual. Alginate, Pel and PsI are extracellular polysaccharides that make up most of the extracellular matrix. There are also proteins such as CdraA and extracellular DNA that increase structural strength of the matrix.

  • Line 181 Siala at al [20]

Fixed as recommended in the revised version.

  • Line 279 DIAR methods – please explain this aberration

The term DIAR was defined under “Managing PJI” section, and in the first paragraph in the revised manuscript line 281. The abbreviation DIAR stands for Debridement, Antibiotics therapy and Implant Retention.

  • Line 360.. Figure 3 looks at drug release kinetics that are involved in delivering antibiotics locally to fight infection- Should be Figure 2???

We have implemented the recommended edit in the revised manuscript on the second paragraph of page 8. The number of the figure was corrected from figure 3 to figure 2 to refer to the correct figure.

  • Figure 2 On the Y-axis there should be some values, eg µg/ml or similar, because the amount of released antibiotic has a unit?

Yes, we added the unit for the figure in the revised manuscript. It is now included in the caption as µg/ml

  • Line 435 CaSO4* 1/2H2O

We have implemented the recommendation to the manuscript. Page 11, second paragraph was edited to include the name of the material, calcium sulfate hemihydrate as well as the formula that represents the hemihydrate CaSO4.1/2H2O

because it is less porous-  It is usually made into pellets that have a high surface area and high porosity to allow for a more consistent release of antibiotics- in the end, alpha has less or more pores? According to the literature, Alpha mixes with less water, so it has fewer pores. It is different if the material is prepared by the manufacturer as a ready implant, then the production process may include the phase of formation of synthetic pores, e.g. by evaporating the substance in gaseous form from the material

Thank you for the suggestion! After reviewing we have accepted the suggestion and made some changes to the manuscript to reflect your suggestion. Specifically, in the second paragraph of page 11, we have updated the text to indicate that the material will have fewer pores. Additionally, we have removed the sentence that stated the hemihydrate form used had high porosity.

  • You can also consider PMMA-based cement, which releases ions from bioactive glasses, e.g.

Raszewski Z, Chojnacka K, Mikulewicz M. Preparation and characterization of acrylic resins with bioactive glasses. Sci Rep. 2022 Oct 5;12(1):16624. doi: 10.1038/s41598-022-20840-1. PMID: 36198737; PMCID: PMC9534886.

On page 10 we discuss the use of PMMA and its history in the management of prosthetic joint infections. Although there are studies that explore the application of PMMA-bioactive glass, the problem lies in the biodegradability of PMMA. Bioactive glass has a lot of potential due to its ability to load drug and its bioactivity which would allow it to degrade and bond to bone.

  • Line 536 The sol gel method or chemistry-based synthesis requires adding the raw material and creating a solution (“sol”) by adding an acid catalyst to the raw material and water- This needs some clarification, we don't add Na2O or SiO2 puree to the solution and expect it to form a glass. Organic compounds are added for the teraethoxy silane precursor Si and others. There is no gel process, only hydrolysis, as a result of which particles are formed that are insoluble in the reaction medium (mixtures of water and alcohol silanes)

We have implemented the recommended edit in the revised manuscript on page 13, second paragraph.

  • Figure 8 gives a, b and c but you have to explain what that means.

Thank you, The suggested revision has been applied to the revised manuscript. Additionally, we have provided an explanation in the caption of Figure 8 to clarify the meaning of (a), (b), and (c). These letters correspond to the different concentrations of gentamicin solutions that were used in the research, specifically, (a) 400 µg/ml, (b) 600 µg/ml, and (c) 800µg/ml.

Reviewer 2 Report

The paper undertook the Mesoporous Bioactive Glass as a Potential Treatment Option for Prosthetic Joint Infections. I did not find the novelty in the present study. The poor presentation and lack of literatures prevent me to accept this article.

1. Please discuss the novelty of the present study. 

2. More literatures are required in term of table.

3. More figures should be incorporate. 

4. Authors need to discuss about biocompatibility. 

5. Authors need to discuss mode of action. 

Author Response

MDPI Branch Office, Belgrade
Bulevar Milutina Milankovica 7v,

11070 Belgrade, Serbia

Re: Revised Manuscript pharmaceutics-2293211- special Issue “"Design of Mesoporous Materials for Biomedical Application"

Dear Editor,

We are pleased to submit our revised manuscript: “Mesoporous Bioactive Glass as a Potential Treatment Option for Prosthetic Joint Infections” by Dana Almasri and Yaser Dahman*. In the revised manuscript we carefully incorporated all changes suggested by the reviewers. In addition, we are pleased to respond point –by– point to the main comments made by the reviewers in the “Detailed Response to Reviewers” file attached.

Best regards,

Yaser Dahman, Ph.D., MBA, P.Eng

Professor, Department of Chemical Engineering

Toronto Metropolitan University

350 Victoria St. Toronto, Ontario M5B 2K3

Phone: 416-979-5000, ext. 554080

Fax: 416-979-5083

Email: ydahman@torontomu.ca

Reviewer 2

  1. Please discuss the novelty of the present study.

The review article attempts to bridge the gap between medical science and materials science by connecting major medical issues with their corresponding solutions. Specifically, the manuscript outlines the problem of prosthetic joint infections and proposes a solution based on a thorough investigation. The study focuses on the applicability of mesoporous bioactive glass as an alternative treatment to current practices. The novelty of the study lies in its unique approach to addressing the problem of prosthetic joint infections by proposing a potential solution using mesoporous bioactive glass. This approach is clearly stated in the abstract, which highlights the objective of the study.

  1. More literatures are required in term of table.

More than 200 articles were critically reviewed and cited in the present review articles to highlight differences between the different bioactive glasses in terms of ability to load drugs and ability to release loaded drugs. Table 3 aims to highlight how different compositions will impact loading and release rate of the drugs. The table looks at MBG loaded with different drugs and with different surface modifications. Accordingly, we don’t feel that more literature citation is required.

  1. More figures should be incorporate. 

The present review has nine figures of high quality. Figures are carefully chosen to help the reader follow along and visualize different concepts. The figures showed biofilm growth stages, drug release kinetics and bioactive glass reactions. As well as looking at specific studies and highlighting necessary figures that can help showcase bioactive glass and its drug loading ability and release kinetics.

  1. Authors need to discuss about biocompatibility. 

Biocompatibility was discussed in the review article under the topic of bioactivity. We agree with the reviewer comments that a clear definition of biocompatibility of bioglass was missing, in addition to discussion on biocompatibility of the bioglass. In the revised manuscript, we have addressed this issue by providing a clear definition of biocompatibility and explaining why bioactive glass is considered biocompatible. We also stated that biocompatibility is defined as “the ability of a material to perform with an appropriate host response” [71] . Bioactive glass is considered biocompatible because it can integrate with the bone in the body and stimulate bone growth, making it both biocompatible and bioactive [72]. Hench (2014) According to Hench (2014), the biocompatibility of bioglass has been demonstrated through in vitro and in vivo studies, which have been approved by the FDA ([75].

In the revised manuscript, we also included a reference that discusses the biocompatibility of bioactive glass

 [72] Maximov, M.; Maximov, O.-C.; Craciun, L.; Ficai, D.; Ficai, A.; Andronescu, E. Bioactive glass—an extensive study of the preparation and coating methods. Coatings 2021, 11 (11), 1386 DOI: 10.3390/coatings11111386. 

  1. Authors need to discuss mode of action. 

 The mode of action of the bioactive glass has already been discussed in detail on page 12. On this page, we highlighted how different compositions change the bioactivity of the glass. We also reviewed how different surface modifications impacts the loading ability of both bioactive glass and mesoporous bioactive glass.

Reviewer 3 Report

In this review, the discussion of mesoporous bioactive glass for prosthetic joint infections is presented. The manuscript needs major revision before it is accepted for publication. 

Major comments

1. The main focus of the review is discussion of the mesoporous bioactive glass and the application for PJI. However, some parts of the review are discussion of biofilm formation and infection treatment, drug loading and release for joint infection treatment. Please consider to change the article title.  

2. Ion doped MBG with antibacterial properties for PJI was not discussed in details.

3. Line 834. Actually, there are several in vivo studies reported. Some examples:

a. Wang, Y.; Pan, H.; Chen, X. The preparation of hollow mesoporous bioglass nanoparticles with excellent drug delivery capacity for bone tissue regeneration. Front. Chem. 2019, 7, 283.

b. Sui, B.; Zhong, G.; Sun, J. Drug-loadable Mesoporous Bioactive Glass Nanospheres: Biodistribution, Clearance, BRL Cellular Location and Systemic Risk Assessment via 45Ca Labelling and Histological Analysis. Sci. Rep. 2016, 6, 33443.

Please update the references and re-write this part.

 Minor comments:

1. Please check if Tables 1, 2 and 3 are referred to in the text.  

2. Line 360, “Figure 3 looks at drug release …..:. Please check if it is Fig. 3.

3. The antibacterial property of the bioactive glass is due to the local increase in pH. However, the local increase in pH may also cytotoxic to “normal” tissue cells.

4. Line 639: “The study found that the costs associated with PJI decreased when bioactive glass because of “decrease in hospital stay, less surgeries involved, and lower antibiotics cost” [82].”. Please check the English grammar.  

5. Fig. 5, reference number is missed.

6. Line 796: “…… shown in figure 8 [93].”. Please check that Fig. 8 should be Fig. 9 and should be another figure.

Author Response

MDPI Branch Office, Belgrade
Bulevar Milutina Milankovica 7v,

11070 Belgrade, Serbia

Re: Revised Manuscript pharmaceutics-2293211- special Issue “"Design of Mesoporous Materials for Biomedical Application"

Dear Editor,

We are pleased to submit our revised manuscript: “Mesoporous Bioactive Glass as a Potential Treatment Option for Prosthetic Joint Infections” by Dana Almasri and Yaser Dahman*. In the revised manuscript we carefully incorporated all changes suggested by the reviewers. In addition, we are pleased to respond point –by– point to the main comments made by the reviewers in the “Detailed Response to Reviewers” file attached.

Best regards,

Yaser Dahman, Ph.D., MBA, P.Eng

Professor, Department of Chemical Engineering

Toronto Metropolitan University

350 Victoria St. Toronto, Ontario M5B 2K3

Phone: 416-979-5000, ext. 554080

Fax: 416-979-5083

Email: ydahman@torontomu.ca

Reviewer 3

Major comments

  1. The main focus of the review is discussion of the mesoporous bioactive glass and the application for PJI. However, some parts of the review are discussion of biofilm formation and infection treatment, drug loading and release for joint infection treatment. Please consider to change the article title.  

This review aims to give the reader an overview of prosthetic joint infections, difficulty in treating a biofilm infection, currently used treatment methods and to also highlight the need for a new biomaterial that can be used in the future. We believe that the title of “Mesoporous Bioactive Glass as a Potential Treatment Option for Prosthetic Joint Infections” provides a wider window that covers all kind of aspects of PJIs. Alternatively, we can rename the review as “Prosthetic Joint Infections: Biofilm Formation, Management, and the Potential of Mesoporous Bioactive Glass as a Treatment Option - A Comprehensive Review”

  1. Ion doped MBG with antibacterial properties for PJI was not discussed in details.

Thank you for the suggestion! This is a slightly different topic that will merit its own review, we were hoping to cover drug loading in MBG before addressing and reflecting on ion doped MBGs. Ion doped MBG is briefly mentioned on page 21, in the paragraph under the figure

  1. Line 834. Actually, there are several in vivo studies reported. Some examples:
  2. Wang, Y.; Pan, H.; Chen, X. The preparation of hollow mesoporous bioglass nanoparticles with excellent drug delivery capacity for bone tissue regeneration. Front. Chem. 2019, 7, 283.
  3. Sui, B.; Zhong, G.; Sun, J. Drug-loadable Mesoporous Bioactive Glass Nanospheres: Biodistribution, Clearance, BRL Cellular Location and Systemic Risk Assessment via 45Ca Labelling and Histological Analysis. Sci. Rep. 2016, 6, 33443.

Thank you for your suggestions! We agree with the reviewer’s comment and have added a few studies to reflect the work that was done to test MBG in vivo on the second paragraph of page 21

Minor comments:

  1. Please check if Tables 1, 2 and 3 are referred to in the text.  

Thank you, we have implemented the recommended edit in the revised manuscript. Specifically, Table 1 is now cited on page 8, Table 2 is referenced in the second paragraph of page 11, Table 3 is mentioned on page 13, second paragraph, and Table 4 is cited in the second paragraph of page 22.

  1. Line 360, “Figure 3 looks at drug release …..:. Please check if it is Fig. 3.

Thank you, we have implemented the recommended edit in the revised manuscript. The sentence should refer to figure 2 instead of 3.

  1. The antibacterial property of the bioactive glass is due to the local increase in pH. However, the local increase in pH may also cytotoxic to “normal” tissue cells.

Thank you, we have investigated the impact of elevated local pH on healthy tissue cells. however, there is currently insufficient data to substantiate any effect of the localized increase in pH on normal cells.

  1. Line 639: “The study found that the costs associated with PJI decreased when bioactive glass because of “decrease in hospital stay, less surgeries involved, and lower antibiotics cost” [82].”. Please check the English grammar.  

Thank you, we have implemented the recommended edit in the revised manuscript

Fig. 5, reference number is missed.

Thank you

  1. Line 796: “…… shown in figure 8 [93].”. Please check that Fig. 8 should be Fig. 9 and should be another figure.

The figure was incorrectly referenced in the original manuscript, in the revised manuscript, the correct figure was referenced on page 20 first paragraph.

Round 2

Reviewer 2 Report

Accept

Author Response

Thank you for accepting the revised manuscript.

Reviewer 3 Report

Some minor comments:

1. The alternative title is more informative. I suggest to make a little change "Prosthetic Joint Infections: Biofilm Formation, Management, and the Potential of Mesoporous Bioactive Glass as a new Treatment Option - A Comprehensive Review”. 

2. Line 449, "Table 2 discssses ....". Please check the spelling.

3. Line 841, ".... by incorporating different molecules like zinc or magnesium .....". ".... by incorporating different metal ions like zinc or magnesium ....". 

4. Line 851, "..... MBGs have not been fully investigated in vivo.". Then, you discuss a few in vivo studies. This may confuse some readers. You may need to explain it. 

5. Fig. 5, "Figure 5. Formation of micelle as the concentration of surfactant passes the critical micelle concentration (cmc) [?]. A reference number is missed. 

Author Response

Dear Editor,

We are pleased to submit our revised manuscript: “Prosthetic Joint Infections: Biofilm Formation, Management, and the Potential of Mesoporous Bioactive Glass as a new Treatment Option” by Dana Almasri and Yaser Dahman*. In the revised manuscript we carefully incorporated all changes suggested by the reviewers. In addition, we are pleased to respond point –by– point to the main comments made by the reviewers in the “Detailed Response to Reviewers” file attached.

Best regards,

Yaser Dahman, Ph.D., MBA, P.Eng

Professor, Department of Chemical Engineering

Toronto Metropolitan University

350 Victoria St. Toronto, Ontario M5B 2K3

Phone: 416-979-5000, ext. 554080

Fax: 416-979-5083

Email: ydahman@torontomu.ca

Reviewer 3

  1. The alternative title is more informative. I suggest to make a little change "Prosthetic Joint Infections: Biofilm Formation, Management, and the Potential of Mesoporous Bioactive Glass as a new Treatment Option - A Comprehensive Review”. 

The title was changed based on the recommendation. The title of the article is now “Prosthetic Joint Infections: Biofilm Formation, Management, and the Potential of Mesoporous Bioactive Glass as a new Treatment Option”. This change was implemented in the revised manuscript.

  1. Line 449, "Table 2 discusses ....". Please check the spelling.

The spelling of the word was revised and is now spelled as “discusses”. The change is reflected on line 454 in the revised manuscript.

  1. Line 841, ".... by incorporating different molecules like zinc or magnesium .....". ".... by incorporating different metal ions like zinc or magnesium ....". 

The words molecules were changed into “metal ions” based on the suggestions. The change is reflected on line 847 on page 21.

  1. Line 851, "..... MBGs have not been fully investigated in vivo.". Then, you discuss a few in vivo studies. This may confuse some readers. You may need to explain it. 

The paragraph at the end of page 21 was revised to help the readers understand it better. Instead of listing testing in vivo as a disadvantage, we discuss how preliminary studies show a lot of potential for MBG and how further studies are still required to fully understand the MBG performance as an antibiotics carrier and any limitations to its use.

  1. 5, "Figure 5. Formation of micelle as the concentration of surfactant passes the critical micelle concentration (cmc) [?]. A reference number is missed. 

The reference for figure 5 was added in the caption. The figure was reproduced with permission under creative commons, and it serves to help the reader understand how micelles are formed and how they form pores in mesoporous bioactive glass.